# S⁶-DAMON: UNLOCKING STRUCTURED SPARSITY IN SELF-SUPERVISED SPEECH MODELS VIA DATA-MODEL CO-COMPRESSION

## ABSTRACT

Driven by the increasing demand for deploying deep neural network (DNN)-powered automatic speech recognition (ASR) systems on mobile platforms, speech models pretrained through self-supervised learning (SSL) have emerged to reduce reliance on the availability of transcribed speech data. However, this has enlarged the gap between the prohibitive model complexity and the limited resources of mobile devices. Therefore, there is a strong desire to streamline the complexity of speech SSL models for real-time acceleration on mobile platforms, which is particularly challenging as the pretrained speech representation may undergo significant degradation. To this end, we develop a framework dubbed S⁶-DAMON to unlock **s**tructured **s**parsity in **s**peech **SSL** models via **da**ta-**mo**del co-compressio**n**. On the data side, leveraging both the duration of each phoneme and the pauses between phonemes of human utterances, we develop a **sal**ient **a**udio token **d**etector, dubbed SALAD, to remove redundant input audio tokens; On the model side, we identify that the failure of SOTA ASR pruning methods under structured sparsity is caused by a sparsity discrepancy between finetuning/deployment and their limited adaptability of sparsity distributions. We address this through a new ASR pruning pipeline named SAFARI, which adopts a three-step pipeline - **s**parsify, **f**inetune, and **a**djust spa**r**s**i**ty. Extensive experiments validate that S⁶-DAMON can significantly accelerate speech SSL models on mobile devices with limited transcribed speech data while maintaining decent ASR accuracy. All source code will be released.

## 1 INTRODUCTION

Recent breakthroughs in deep neural networks (DNNs) have tremendously advanced the field of Automatic Speech Recognition (ASR), enabling record-breaking end-to-end ASR systems (Hannun et al., 2014; Zhang et al., 2020; Gulati et al., 2020). Considering that speech is one of the basic input modalities of intelligent mobile devices, there has been an increasing interest in the development and deployment of on-device ASR systems.

Despite the big promise, there still remain two critical efficiency bottlenecks for ubiquitous on-device ASR systems, including (1) *data efficiency:* big data is often impractical for ASR since collecting transcription on a large scale is costly or may not be even possible, and (2) *model efficiency:* the often limited on-device resources stand at odds with the complexity of large ASR models. To promote the aforementioned data efficiency, recent advances in self-supervised learning (SSL) for speech representation (Baevski et al., 2020; 2022) have demonstrated empirical success and become the de-facto paradigm for low-resource ASR. However, this could further aggravate the model efficiency bottleneck as large transformers (Vaswani et al., 2017) are often adopted in state-of-the-art (SOTA) speech SSL models to ensure effective representation learning, making it increasingly more challenging for on-device deployment. Therefore, it is imperative to compress speech SSL models while maintaining their generalizable speech representation for delivering efficient ASR systems.

Despite the demand for efficient ASR systems, it is non-trivial to narrow the gap between large speech SSL models and constrained resources in mobile devices. First, under the SOTA pretrain-and-finetune paradigm, the most useful features are learned during the SSL stage and it is difficult to induce sparsity during finetuning while still preserving the fidelity of the speech representation *given the low-resource transcribe speech*. Second, unstructured sparsity induced at the granularity of

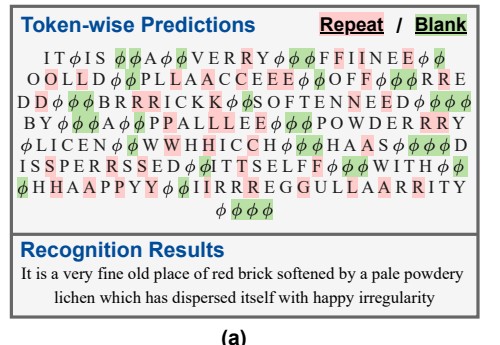 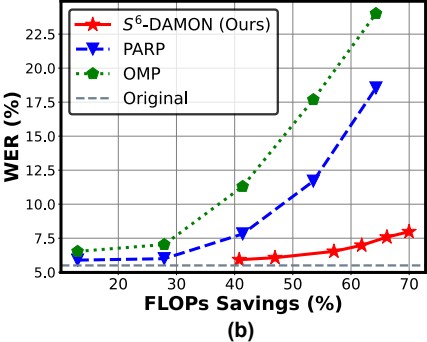

(a) (b)

Figure 1: (a) An example from LibriSpeech (Panayotov et al., 2015) for illustrating two types of non-salient audio tokens; (b) The trade-offs between WER and FLOPs savings achieved by our $S^6$-DAMON and SOTA ASR compression schemes on wav2vec2-base finetuned on LibriSpeech-100h.

weight elements cannot be effectively utilized by commercial mobile devices. Conversely, inducing structured sparsity (Wen et al., 2016), which eliminates all computations associated with a set of neurons, resulting in real-device acceleration, will significantly disrupt the SSL speech representation learned during pretraining. For example, enforcing structured sparsity using the SOTA ASR pruning framework called PARP (Lai et al., 2021) leads to a >8% increase in word-error-rate (WER) with only a 20% sparsity on wav2vec2-base finetuned on LibriSpeech-1h.

**Our contributions.** To tackle this, we develop a framework dubbed $S^6$-DAMON, which for the first time unlocks **s**tructured **s**parsity in **s**peech **SS**L models under low-resource settings through **da**ta-**mo**del co-compressio**n** for enabling real-time on-device speech recognition on mobile platforms.

**On the data side**, $S^6$-DAMON exploits the intrinsic redundancy in human speech to effectively eliminate redundant input tokens and their associated computations. This introduces a new granularity of structured sparsity at the data level, complementing the existing model sparsity. By considering the duration of each phoneme and the pauses in human utterances, we observe that the sampled audio frames and their corresponding extracted audio tokens, which serve as inputs to the transformers, may either (1) repeat previous tokens or (2) be empty, contributing little to the final recognition (refer to an example in Fig. 1 (a)). We refer to these tokens as *non-salient audio tokens (NATs)*, and term the first-appearing tokens that are indispensable for ensuring monotonic recognition as *salient audio tokens (SATs)*. Since properly removing NATs can lead to significant gains in model efficiency while maintaining accuracy better than removing SATs, we develop a **sal**ient **a**udio token **d**etector called SALAD to detect and effectively remove NATs. Given the absence of token-wise labels in ASR datasets for classifying SATs/NATs, our SALAD is trained in a semi-supervised manner based on token-wise pseudo labels provided by a finetuned speech SSL model on untranscribed speech.

**On the model side**, we observe that the failures of SOTA ASR pruning methods under structured sparsity stem from (1) the discrepancy in sparsity distributions between finetuning and deployment. For instance, PARP (Lai et al., 2021) iteratively restores the pruned weights to non-zero values during finetuning to flexibly adapt sparsity distributions, which leads to a discrepancy against the hard-pruned weights during deployment; (2) the limited adaptability of sparsity distributions due to the intrinsically low learning rates during finetuning, resulting in an under-exploration of the space of sparsity masks. Hence, we hypothesize that the key to enabling structured sparsity on speech SSL models is to *ensure the flexibility of adjusting the sparsity masks while avoiding the discrepancy of sparsity distributions between finetuning and final deployment*. We embody this insight in a new ASR pruning pipeline named SAFARI (i.e., **s**p**a**rsify, **f**inetune, and **a**djust sp**a**r**si**ty) to strictly zero-out the pruned neurons during finetuning, thereby minimizing the aforementioned discrepancy. This is followed by a sparsity adjustment step to adaptively evolve the sparsity masks and ensure the adaptability of sparsity distributions. We summarize our contributions as follows:

- We develop a data-model co-compression framework, dubbed $S^6$-DAMON, which for the first time unlocks structured sparsity in both input audio tokens and model structures of speech SSL models to empower real-time on-device ASR under a low-resource setting;

- We propose a semi-supervised method for training a lightweight module dubbed SALAD to distinguish SATs/NATs for the purpose of structurally removing redundant audio tokens;

- We identify the underlying causes for the failures of SOTA ASR pruning methods and subsequently develop an ASR pruning pipeline dubbed SAFARI to enable high structured sparsity in speech SSL models while maximally maintaining the ASR accuracy;

- Extensive experiments and on-device measurements show that as compared to the SOTA ASR pruning method PARP (Lai et al., 2021), our $S^6$-DAMON can (1) achieve a 1.96× speed-up on a Pixel 3 phone with an absolute 2.49% WER reduction, and (2) win a 10.96% lower WER for saving >64% floating-point operations (FLOPs) as shown in Fig. 1 (b).

We emphasize that considering the prevalence of foundation models, the growing ambition for their on-device deployment necessitates new customized compression paradigms to preserve their pretrained representation. Our work represents one of the early efforts towards this objective, and the insights we provide may serve as a guiding beacon for future innovations in other modalities.

## 2 RELATED WORK

**Automatic speech recognition.** Early ASR systems (Sha & Saul, 2006; Tang, 2009; Adams & Beling, 2019) mainly build on top of the combinations of hidden Markov models with Gaussian mixture models or DNNs, and often integrate multiple modules, e.g., an acoustic model, a language model, and a lexicon model, which are separately trained. Driven by recent advances in DNN structures, diverse end-to-end ASR systems have been proposed, including CTC-based models (Graves et al., 2006; Hannun et al., 2014; Amodei et al., 2016), recurrent neural network (RNN)-transducers (Graves, 2012; Graves et al., 2013; Dong et al., 2018b), and sequence-to-sequence models (Chorowski et al., 2015; Chan et al., 2016). Specifically, from the model structure perspective, transformer-based models (Gulati et al., 2020; Dong et al., 2018a; Wang et al., 2020) have been widely adopted thanks to their superior expressiveness and capabilities for modeling long-range dependencies.

**Self-supervised learning for speech representation.** To learn rich speech representation via SSL, early works design generative models for inferring the latent variables of speech units (Hsu et al., 2017; van den Oord et al., 2017; Khurana et al., 2020). Recently, prediction-based SSL methods have gained more attention, where the models are trained to reconstruct the contents of unseen frames (Chung et al., 2019; Chi et al., 2020; Baevski et al., 2022) or contrast the features of masked frames with those of randomly sampled ones (Baevski et al., 2020; Conneau et al., 2020; Hsu et al., 2021). In parallel, some works combine both predictive and contrastive objectives (Baevski et al., 2019b;a) or integrate contrastive learning and masked language modeling (Chung et al., 2021; Bapna et al., 2022). We refer the readers to the survey (Liu et al., 2022) for more details. However, SOTA speech SSL models (Baevski et al., 2020; Hsu et al., 2021; Baevski et al., 2022; Conneau et al., 2020) often adopt large transformers for ensuring effective representation learning, making it difficult to achieve real-time speech recognition on mobile devices.

**ASR pruning.** To compress large-scale ASR models while maintaining their generalizable representation, ASR pruning has gained growing attention. Early works prune either the decoding search space (Pylkkönen, 2005; Xu et al., 2018; Zhang et al., 2021) or the HMM state space (Van Hamme & Van Aelten, 1996). Recent works have shifted their focus to pruning end-to-end ASR models (Venkatesh et al., 2021; Shi et al., 2021; Li et al., 2021b). Recently, (Lai et al., 2021; Prasad et al., 2022; Zhao et al., 2021) prune speech SSL models towards more efficient low-resource ASR, however, they all adopt unstructured pruning which barely favors hardware efficiency in commercial devices. (Lee et al., 2022; Chang et al., 2022) distill the knowledge of pretrained speech SSL models to lightweight student models but require human expertise to manually design the student model, causing inferior ASR accuracy without utilizing the intrinsic properties of speech signals. In contrast, our $S^6$-DAMON learns to *automatically* and *structurally* prune the redundancy of speech SSL models, achieving a triple-win in data, model, and labor efficiency.

## 3 THE PROPOSED $S^6$-DAMON FRAMEWORK

### 3.1 FRAMEWORK OVERVIEW

**Rationale.** Structured sparsity can yield more substantial acceleration for commercial mobile devices compared to unstructured sparsity. However, the learned representation of speech SSL models may experience significant degradation, especially when finetuned in a low-resource setting. In addressing this, our $S^6$-DAMON approaches this challenge with a twofold rationale: (1) instead of compressing

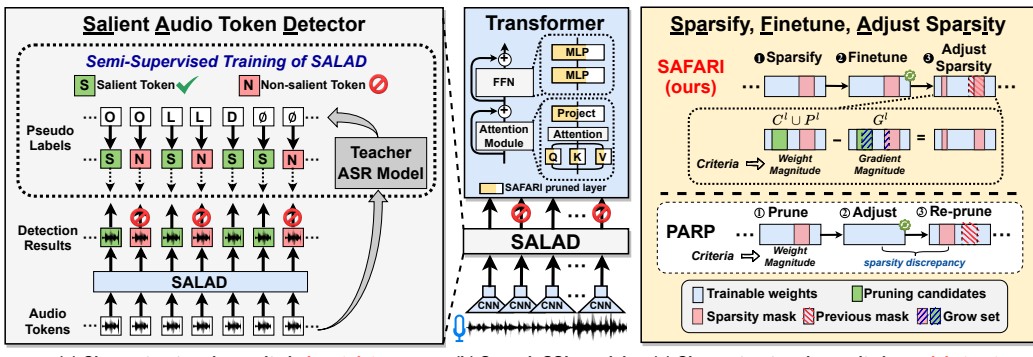

Figure 2: An overview of our proposed $S^6$-DAMON, integrating two enabling components called (a) SALAD and (c) SAFARI for unlocking structured sparsity in input data and model structures, respectively, of (b) speech SSL models (zoom-in for better view).

only one dimension, $S^6$-DAMON exploits redundancy in both the input audio tokens and model structures; and (2) to preserve the fidelity of speech representation, $S^6$-DAMON first identifies and then addresses the key bottlenecks for reducing the input data and model redundancy, respectively.

**Framework overview.** As shown in Fig. 2, $S^6$-DAMON performs data-model co-compression via (1) a SALAD module which detects and skips redundant tokens, i.e., NATs as introduced in Sec. 1, and (2) a SAFARI pipeline which pursues structured model sparsity. For compressing a given speech SSL model, $S^6$-DAMON features a three-stage pipeline: ❶ finetune the speech SSL model that serves as a teacher model on the low-resource transcribed speech; ❷ train SALAD in a semi-supervised manner based on the annotations of the teacher model (see Sec. 3.2); and ❸ perform a joint optimization of input data sparsity and model sparsity based on the SAFARI pipeline with a portion of detected NATs removed (see Sec. 3.3 and Sec. 3.4).

## 3.2 SALAD: CHASING STRUCTURED SPARSITY IN INPUT DATA

In this subsection, we present the rationale, design, and training of our SALAD module, which aims to detect and skip redundant input audio tokens, referred to as NATs as introduced in Sec. 1.

**SALAD's rationale.** The objective of our SALAD is to explicitly harness redundancy in repeated/blank regions of human utterances. At a high level, SALAD achieves this by distilling knowledge about token-wise redundancy from a finetuned speech SSL model to our target model using a lightweight detector as a bridge in a semi-supervised learning pipeline. This differs from previous methods (Bhati et al., 2021; Afshan & Alwan, 2022) that achieve variable-frame-rate ASR through re-sampling based on entropy or frame-wise dissimilarity. Our approach, for the first time, focuses on SOTA speech SSL models built on large transformers and utilizes their outputs as guidance to detect repeated/blank regions of human utterances from the models' perspective, rather than solely relying on input speech statistics.

**SALAD's input and structure.** SOTA speech SSL models (Baevski et al., 2020; Hsu et al., 2021; Baevski et al., 2022) sample and convert raw audio frames into audio tokens via a convolutional feature extractor, which are then processed by a transformer backbone to generate corresponding contextualized representation. As the convolutional feature extractor is often fixed after SSL pretraining to ensure effective audio feature extraction (Baevski et al., 2020), SALAD is applied to the extracted audio tokens after the convolutional feature extractor and classifies each audio token as an SAT/NAT. This ensures that all transformer layers can benefit from reducing the same number of audio tokens. Specifically, SALAD consists of four lightweight convolutional layers, accounting for $<0.4\%$ FLOPs of the original transformer, and outputs a binary classification between SAT and NATs.

**SALAD's semi-supervised learning pipeline.** Considering the lack of token-wise ground truth, we train $SALAD(\cdot; \theta_S)$ in a semi-supervised manner, i.e., using a finetuned ASR model to provide pseudo labels for each token on untranscribed speech (see Fig. 2 (a)). Specifically, given a speech SSL model, we first finetune it on the available transcribed speech (e.g., LibriSpeech-1h) to create a teacher model $M_T(\cdot; \theta_T)$, which is then used to annotate a larger amount of untranscribed speech (e.g., LibriSpeech-10h) to acquire the pseudo labels for each audio token. Correspondingly, the binary pseudo labels of SATs/NATs can be derived for each audio token based on whether it repeats the

previous token or is a blank token. Note that although we adopt CTC (Graves et al., 2006) as the label topology as an illustration in Fig. 1 (a) and Fig. 2 (a), which is a common practice of speech SSL models (Baevski et al., 2020; Conneau et al., 2020; Hsu et al., 2021; Baevski et al., 2022), our SALAD does not rely on specific label topology as it leverages the intrinsic redundancy in human utterances, i.e., the duration of each phoneme and the pauses between the phonemes.

Enforce a high recall on SATs. One issue is that the consequence of classifying an SAT to a NAT is severe, under which SATs are more likely to be mistakenly skipped in inference, harming the monotonic input/output alignment. It is thus highly desired to maximize the coverage of SATs (i.e., a high recall on SATs) during training SALAD. To achieve this, we exert a larger penalty when an SAT is misclassified as a NAT, which can be formulated as:

$$\arg\max_{\theta_S} \sum_{i=1}^{t} \alpha_i L(SALAD(x_i; \theta_S), Bin(M_T(x_i; \theta_T))) \quad (1)$$

where $L$ is a binary cross-entropy loss, $x_i$ is the $i$-th audio token extracted by the convolutional feature extractor, $Bin$ denotes a transformation from pseudo character/phoneme labels to binary labels of SATs/NATs, and $\alpha_i$ is a penalty coefficient for enforcing high recalls on SATs.

**Speech SSL model finetuning with SALAD.** Although NATs are less likely to impact the monotonic input/output alignment as compared to SATs, removing NATs could result in domain gaps in terms of speech speed and rhythm between pretraining and finetuning. As such, a finetuning process is required to fill in the domain gap. Specifically, we finetune the target speech SSL model integrated with SALAD, where a certain ratio of detected NATs is removed before being fed into the transformer.

Implementation. To balance ASR accuracy and efficiency, we set a skip ratio $sr$ for NATs in all the input audio when finetuning with SALAD, i.e., for the NATs detected by SALAD in an input audio clip, we remove the top $sr$ NATs sorted in terms of confidence score predicted by SALAD, and the remaining NATs and all detected SATs are then fed into the transformer. Since different audio clips contain different percentages of NATs, which can cause different audio token lengths for samples in a batch, we pad each sample to the largest token length of each batch during finetuning.

### 3.3 SAFARI: CHASING STRUCTURED SPARSITY IN MODEL STRUCTURES

**Rethink the SOTA ASR pruning method.** The core concept behind the SOTA ASR pruning method PARP (Lai et al., 2021) lies in enabling gradient propagation to pruned weights during finetuning, thereby allowing for the update of sparsity distributions. This proves to be crucial for sparsifying speech SSL models, as the weight magnitudes inherited from SSL pretraining may not accurately indicate the importance of neurons for downstream tasks. The significance of such adaptable sparsity distributions is confirmed in PARP through *unstructured* pruning, in comparison to one-shot/iterative magnitude pruning (OMP/IMP) (Lai et al., 2021).

**Identified issues of the SOTA ASR pruning method.** Directly extending PARP to a structured pruning setting will cause a failure. We extend PARP's setting to structured sparsity, i.e., remove all connections from the pruned input neurons, for pruning wav2vec2-base on LibriSpeech-10m/1h under a 20% sparsity ratio and vary the pruning intervals in terms of itera-

Table 1: Apply OMP/PARP with varied pruning intervals on top of wav2vec2-base when being finetuned on LibriSpeech-10m/1h under 20% sparsity.

| Dataset | Method | Learn-ability | No Dis-crepancy | Pruning Interval | | | | |
|---------|--------|---------------|-----------------|---|---|---|---|---|
| | | | | 1 | 2 | 5 | 10 | 50 |
| Libri-10m | PARP | ✔ | ✗ | **53.24** | 54.34 | 57.47 | 65.85 | 87.65 |
| | OMP | ✗ | ✔ | | | 58.75 | | |
| Libri-1h | PARP | ✔ | ✗ | **26.90** | 27.30 | 31.18 | 32.77 | 42.89 |
| | OMP | ✗ | ✔ | | | 29.01 | | |

tions between prune/re-prune. Tab. 1 shows that (1) the original pruning interval adopted by PARP (i.e., 50 iterations) leads to an absolute 23.93% WER increase over standard wav2vec2-base (18.96% WER) on LibriSpeech-1h; (2) setting a small pruning interval could lead to reduced WER over OMP; and (3) larger pruning intervals consistently cause more notable performance degradation, where the softly pruned weights diverge more from zero as they can be updated in PARP's adjustment step.

Analysis. This set of experiments indicates that while enhancing the adaptability of sparsity masks in PARP is beneficial, it can lead to a sparsity discrepancy between finetuning (i.e., not exactly zero) and final deployment (i.e., hard pruning), thereby harming the delicate speech representation inherited from SSL pretraining, particularly under structured sparsity. *Furthermore*, the flexibility of adjusting the sparsity masks in PARP remains limited; for instance, only <3% of elements in the sparsity masks are updated throughout the finetuning process, aligning with PARP's observed >99% Intersection

Over Union (IOU) between the initial and final subnetworks. This is attributed to the intrinsically low learning rates during finetuning, resulting in the sparsity distributions being largely under-explored when updated via gradients in PARP.

**The SAFARI pipeline.** The above analysis indicates that the key to pursuing structured sparsity in speech SSL models is to *ensure the adaptability of sparsity distributions while minimizing the sparsity discrepancy between finetuning and deployment*. Thus, we instantiate this insight into a new pruning pipeline SAFARI: ❶ *sparsify*: prune a speech SSL model to the target sparsity $sp$ based on the weight magnitudes; ❷ *finetune*: finetune the model weights with the sparsity mask applied, i.e., the pruned weights are zero-outed without receiving gradients to avoid the sparsity discrepancy; and ❸ *adjust sparsity*: the sparsity mask is adaptively adjusted for boosting the adaptability of sparsity distributions. Steps ❷ and ❸ are iterated towards convergence. SAFARI can be viewed as an intermediate choice between OMP and PARP, marrying the former's stability and the latter's adaptability.

Implementation of SAFARI. There can be different ways to implement the above spirit, i.e., SAFARI. Inspired by (Evci et al., 2020), we adopt gradient magnitudes as a criterion to adjust the sparsity masks in a prune-and-grow manner (see Fig. 2 (c)). Specifically, in each sparsity adjustment step, for a set of neurons $|N^l|$ and a set of pruned neurons $P^l$ ($|P^l|/|N^l| = sp$) in the $l$-th layer, SAFARI ① selects $ar$ neurons from $N^l \setminus P^l$ as the pruning candidate set $C^l$ ($|C^l|/|N^l| = ar$) based on a pruning criterion, where $ar$ is a predefined adjustment ratio, and ② chooses $ar$ neurons from the joint set of pruning candidates and pruned neurons $P^l \cup C^l$ ($= P^l + C^l$) to form a growing set $G^l$, which are allowed to be updated by the gradients of the next finetuning step, based on a growing criterion. Therefore, the new sparsity mask applied in the next finetuning step is built by $P^l + C^l - G^l$, which has a constant sparsity ratio $sp$. More specifically, we adopt the $\ell_1$-norm of the weight vectors from a neuron, i.e., $||W_{i,\cdot}^l||_{\ell_1}$ for the $i$-th neuron, as the pruning criterion (i.e., prune the smallest ones), and the corresponding gradients $||\frac{\partial L}{\partial W_{i,\cdot}^l}||_{\ell_1}$ as the growing criterion (i.e., grow the largest ones).

### 3.4   $S^6$-DAMON: Joint Data-Model Co-Compression

To perform joint optimization of input data sparsity and model sparsity, we integrate the target speech SSL model with SALAD and finetune it via the SAFARI pipeline with a portion of detected NATs removed. To push forward the achievable accuracy-efficiency trade-off, $S^6$-DAMON can optionally enable a semi-supervised distillation mechanism to boost the achievable ASR accuracy. Specifically, we distill the knowledge of the teacher model mentioned in Sec. 3.2 to the compressed model in a layer-wise manner during finetuning on top of a mixed dataset composed of both transcribed speech $D_T$ and untranscribed speech $D_U$, where the pseudo labels on untranscribed speech are annotated by the teacher model. Note that the teacher model is only finetuned on the limited transcribed speech. The objective of the semi-supervised distillation process can be formulated as:

$$\mathcal{L} = \sum_{x \in D} \sum_{l=1}^{L} MSE(h_\theta^l(x), h_{\theta_T}^l(x)) + \sum_{x \in D_T} CTC(h_\theta^L(x), y) \tag{2}$$

where $D = D_T \cup D_U$, $MSE$ and $CTC$ are the loss functions, $h_\theta^l(x) = M^l(SALAD(x; \theta_S); \theta)$ is the hidden representation for the remained tokens in the $l$-th layer of the compressed model $M$, $h_{\theta_T}^l(x) = \mathbb{S} \circ M_T^l(x; \theta_T)$ is the corresponding hidden representation of the teacher model and $\mathbb{S}$ is a selection operator for only calculating the MSE loss on the remained tokens determined by SALAD.

## 4   Experimental Results

### 4.1   Experiment Setup

**Models and datasets.** We evaluate our $S^6$-DAMON on four SOTA speech SSL models, including wav2vec2-base/large (Baevski et al., 2020), data2vec (Baevski et al., 2022), and hubert (Hsu et al., 2021) pretrained on LibriSpeech (Panayotov et al., 2015) in an SSL manner. We evaluate the compression effectiveness under different resource settings, including LibriSpeech-10m/1h/10h/100h/960h following the split in (Baevski et al., 2020). In addition to ASR, we also consider six speech processing tasks from SUPERB (Yang et al., 2021). For results on LibriSpeech, we report the WER on test-clean by default. The detailed finetuning settings are provided in Appendix G.

**$S^6$-DAMON settings:** For SALAD training, we adopt the same training schedule as finetuning the speech SSL model weights and the $\alpha_i$ in Eq. 1 is 10 for penalizing mistakes on SATs otherwise 1.

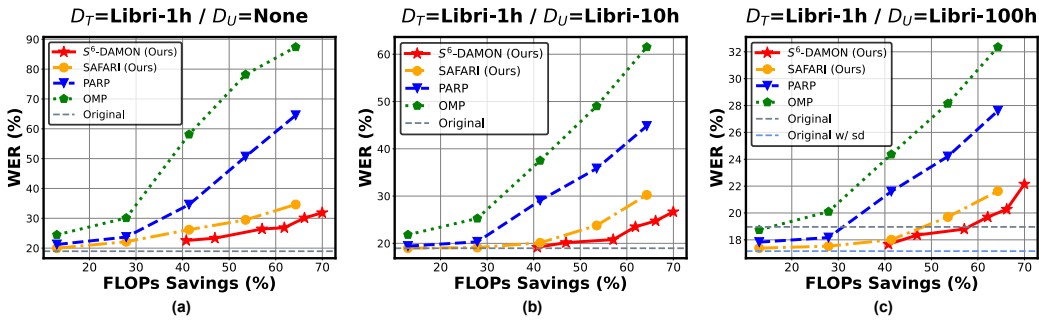

Figure 3: Benchmark our $S^6$-DAMON and SAFARI with SOTA ASR pruning methods PARP and OMP on wav2vec2-base with transcribed LibriSpeech-1h and different untranscribed resources. "w/ sd" denotes applying the semi-supervised distillation for finetuning the original model, which could notably reduce the WER when the untranscribed LibriSpeech-100h is available.

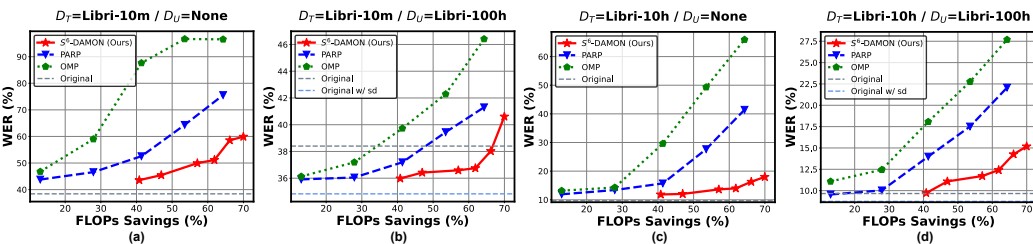

Figure 4: Benchmark our $S^6$-DAMON with PARP and OMP under different low-resource settings.

For SAFARI, we adjust the sparsity every 50 iterations and fix the sparsity mask after 10k iterations for all experiments. By default, the adjustment ratio $ar$ is set the same as the target sparsity $sp$ if not specifically stated, which is justified by the ablation study in Appendix F.2. We adopt a NAT skip ratio $sr$ of 0.4/0.6/0.8 (i.e., $sr$ of NATs are removed) and note that the final ratio of skipped tokens to the total tokens would depend on the number of NATs detected in the given speech. If not specifically stated, we report the FLOPs savings of the transformer component, which often dominates the total FLOPs (e.g., >63% for wav2vec2-base), in speech SSL models.

### 4.2 BENCHMARK WITH SOTA ASR PRUNING METHODS

Considering that feed-forward networks (FFNs) are more sensitive to structured sparsity than self-attention (SA) as demonstrated in Appendix F.1, for both our method and baselines, given a target sparsity $sp$, we by default set their sparsity to satisfy $(sp_{SA} + sp_{FFN})/2 = sp$ and $sp_{SA} - sp_{FFN} = 0.2$, which achieves better ASR accuracy with comparable FLOPs as compared to uniformly setting a sparsity of $sp$. For PARP, we adopt its best-performed setting for structured pruning, i.e., update the sparsity every iteration according to the analysis in Sec. 3.3 and Tab. 1.

**Benchmark on English ASR under different low-resource settings.** We benchmark our $S^6$-DAMON with OMP and PARP (Lai et al., 2021) and apply the semi-supervised distillation described in Eq. 2 to both our method and the baselines for fair comparisons. In particular, for a comprehensive benchmark, we vary the available resources in the transcribed data $D_T$ and those in the untranscribed data $D_U$. We adopt a NAT skip ratio (i.e., $sr$ in Sec. 3.2) of 0.4∼0.8 for our SALAD technique and a sparsity ratio (i.e., $sp$) of 0.2∼0.5 for our SAFARI technique and other ASR pruning baselines.

Observation and analysis. As shown in Fig. 3, we can observe that (1) our $S^6$-DAMON consistently outperforms PARP and OMP by a notable margin, e.g., an absolute >7% WER reduction as compared to PARP for achieving >64% FLOPs savings on wav2vec2-base with LibriSpeech-1h/100h as $D_T/D_U$; (2) our $S^6$-DAMON shows decent scalability under more stringent low-resource settings where PARP/OMP fail to achieve acceptable recognition effectiveness, e.g., an absolute up-to-34% lower WER over PARP when only LibriSpeech-1h is available.

In addition, enabling both SALAD and SAFARI can consistently win a better WER-FLOPs trade-off over SAFARI, especially under more stringent low-resource settings. As shown in Tab. 7, which is an exemplary breakdown summarized

Table 2: Breakdown of the WER reduction.

| Method | PARP (Lai et al., 2021) | SAFARI (Ours) | SAFARI + SALAD (Ours) |
|---|---|---|---|
| FLOPs Savings (%) | 64.29% | 64.29% | **66.18%** |
| $D_T$=Libri-1h + $D_U$=Libri-10h | 64.52 | 34.63 | **30.12** |
| | 44.81 | 30.25 | **24.74** |

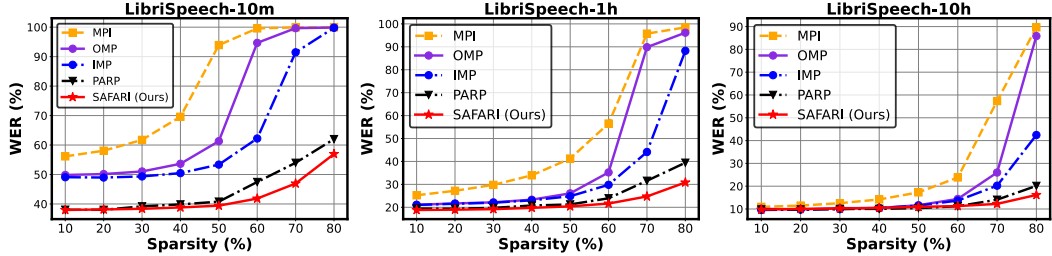

Figure 5: Benchmark our SAFARI with PARP, OMP, IMP, and MPI (reported by (Lai et al., 2021)) for unstructured pruning on wav2vec2-base when being finetuned on LibriSpeech-10m/1h/10h.

from Fig. 3, enabling both SALAD and SAFARI achieves an absolute 4.51%~5.51% lower WER over SAFARI for reducing >64% FLOPs on LibriSpeech-1h. This indicates that input data sparsity is a critical dimension for structurally trimming down the complexity of speech SSL models in addition to model sparsity. A more comprehensive breakdown is provided in Appendix B.

**More low-resource settings.** We further evaluate our $S^6$-DAMON under more low-resource settings, i.e., LibriSpeech-10m/10h w/ and w/o semi-supervised distillation. As shown in Fig. 4, our $S^6$-DAMON consistently outperforms PARP and OMP across all settings. This indicates its decent scalability with resources, which is highly desirable for real-world ASR deployment.

**Benchmark under high-resource settings.** We further evaluate our method given more downstream resources, including LibriSpeech-100h/960h as shown in Fig. 1 (b) and Tab. 3, respectively. We can observe that (1) our $S^6$-DAMON still outperforms the baselines, e.g., an absolute 10.96%/5.92% lower WER over PARP for reducing >50% FLOPs on wav2vec2-base with 100h/960h transcribed data, respectively, and (2) enabling both SAFARI and SALAD again achieves both lower WER and smaller FLOPs than enabling SAFARI only under a high-resource setting according to Tab. 3.

Table 3: Benchmark our method with PARP when being finetuned on LibriSpeech-960h.

| Method | Sparsity | GFLOPs | WER (%) |
|---|---|---|---|
| Original | - | 47.1 | 3.39 |
| PARP (Lai et al., 2021) | $sp$=0.4 | 21.89 (-53.52%) | 10.26 |
| SAFARI (ours) | $sp$=0.4 | 21.89 (-53.52%) | 5.01 |
| S6-DAMON (ours) | $sr$=0.6, $sp$=0.3 | 20.22 (-57.07%) | 4.34 |

**Benchmark under unstructured sparsity.** We validate the scalability of our SAFARI to unstructured sparsity via benchmarking with the reported results of PARP, OMP, IMP, and MPI (i.e., magnitude pruning at pretrained initializations) in (Lai et al., 2021) without any distillation. As shown in Fig. 5, we can observe that our SAFARI can outperform all baseline methods across all resource settings, especially under large sparsity ratios, e.g., an absolute 8.61% lower WER under 80% sparsity over PARP on LibriSpeech-1h. This indicates that (1) the sparsity discrepancy issue still exists in unstructured pruning under a larger sparsity ratio, and (2) our SAFARI pipeline consistently shows its superiority as an ASR pruning paradigm over PARP under both structured/unstructured sparsity.

**Benchmark with distillation-based models.** We benchmark with the reported ASR results in DistilHuBERT (Chang et al., 2022) and FitHu-BERT (Lee et al., 2022) for compressing hubert and wav2vec2-base on LibriSpeech-100h. As shown in Tab. 4, $S^6$-DAMON achieves an absolute 7.04% lower WER with 8.7% fewer param-

Table 4: Benchmark $S^6$-DAMON with distillation-based ASR compression methods (Chang et al., 2022; Lee et al., 2022).

| Method | Model | Params (M) | WER (%) |
|---|---|---|---|
| DistilHuBERT (Chang et al., 2022) | hubert (Hsu et al., 2021) | 23.49 | 13.37 |
| FitHuBERT (Lee et al., 2022) | wav2vec2 (Baevski et al., 2020) | 22.49 | 14.77 |
| | hubert (Hsu et al., 2021) | 22.49 | 12.66 |
| $S^6$-DAMON (Ours) | wav2vec2 (Baevski et al., 2020) | 20.53 | **7.73** |
| | hubert (Hsu et al., 2021) | 20.53 | **7.94** |

eters as compared to the strongest baseline FitHuBERT, indicating that given a speech SSL model, trimming down its complexity in a top-down manner may achieve better compression effectiveness than manually designing an efficient model from scratch.

**Benchmark on more speech SSL models.** We further extend our $S^6$-DAMON to more models, i.e., data2vec (Baevski et al., 2022) and wav2vec2-large (Baevski et al., 2020) on LibriSpeech-1h (i.e., no $D_U$). As shown in Fig. 6, we can observe that (1) our method shows consistent WER reductions over PARP on different speech SSL models, and (2) according to the comparison between wav2vec2-base/large, structurally compressing a larger speech SSL model may not result in better WER-FLOPs trade-offs than compressing a smaller

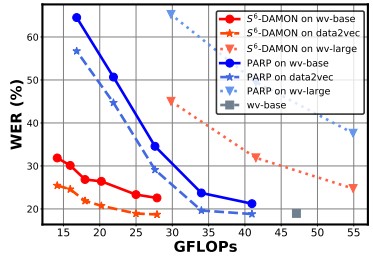

Figure 6: Benchmark with PARP on top of wav2vec2-base/large and data2vec.

Table 5: Benchmark our $S^6$-DAMON with PARP on six tasks from SUPERB (Yang et al., 2021).

| Method | FLOPs Saving (%) | Params Saving (%) | ER (Acc ↑) | KS (Acc ↑) | QbE (MaxTWV ↑) | IC (Acc ↑) | SF (F1 ↑) | ASV (EER ↓) |
|---|---|---|---|---|---|---|---|---|
| Original | 0.00 | 0.00 | 0.626 | 0.962 | 0.053 | 0.966 | **0.874** | **0.061** |
| PARP (Lai et al., 2021) | 53.52 | 57.14 | 0.622 | 0.957 | 0.075 | 0.967 | 0.851 | 0.065 |
| $S^6$-DAMON (Ours) | 57.07 | 44.39 | **0.654** | **0.964** | 0.118 | 0.982 | 0.867 | 0.063 |
| | **69.98** | **57.14** | 0.641 | 0.957 | **0.122** | **0.983** | 0.855 | 0.063 |

Table 6: Measure the latency of compressed models on a Google Pixel 3 mobile phone. "FE" denotes the feature extractor and "Trans." denotes the transformer backbone. All models are finetuned on LibriSpeech-100h and $sr/sp$ are the adopted skip ratio/sparsity, respectively.

| Method | $sr / sp$ | Params (M) | WER (%) | Lat. (ms) Conv FE | Lat. (ms) Trans. | Speed-up on Trans. | Speed-up Overall | RTF ↓ |
|---|---|---|---|---|---|---|---|---|
| Original | - / - | 94.74 | 5.50 | 2270.1 | 6678.5 | 1.00× | 1.00× | 0.895 |
| PARP (Lai et al., 2021) | - / 0.3 | 56.77 | 7.82 | 2270.1 | 4809.4 | 1.39× | 1.26× | 0.708 |
| | - / 0.4 | 45.80 | 11.69 | 2270.1 | 3756.6 | 1.77× | 1.48× | 0.603 |
| $S^6$-DAMON (Ours) | 0.6/0.2 | 69.01 | 6.07 | 2270.1 | 3834.5 | 1.74× | 1.47× | 0.610 |
| | 0.6/0.3 | 56.77 | 6.53 | 2270.1 | 3318.7 | 2.01× | 1.60× | 0.559 |
| | 0.8/0.3 | 56.77 | 6.98 | 2270.1 | 3061.1 | 2.18× | 1.68× | 0.533 |
| | 0.8/0.4 | 45.80 | 7.98 | 2270.1 | 2377.9 | 2.81× | 1.93× | 0.465 |
| | 0.8/0.7 | 20.53 | 9.20 | 2270.1 | 819.1 | 8.15× | 2.90× | 0.309 |

one as aggressively compressing a pretrained model could harm the pretrained speech representation. This set of experiments indicates that, under a low-resource setting, using a smaller speech SSL model with mild structured sparsity is preferable compared to larger models with high sparsity.

### 4.3 EXTENSION TO OTHER SPEECH PROCESSING TASKS

**Benchmark on SUPERB.** Although efficient ASR is our main focus, we also evaluate our method on more speech processing tasks from SUPERB. In particular, we transfer the compressed models with LibriSpeech-1h/100h as $D_T/D_U$ of our $S^6$-DAMON and PARP to perform six speech processing tasks on SUPERB (Yang et al., 2021). As shown in Tab. 5, we observe that (1) our method wins four out of six tasks over the original wav2vec2-base with ≥55.07%/44.39% FLOPs/parameter reductions, and (2) our method consistently outperforms PARP across all the tasks. This indicates that our $S^6$-DAMON can potentially serve as a general compression technique for speech processing.

### 4.4 REAL-DEVICE MEASUREMENT OF $S^6$-DAMON

To validate the real-device efficiency of $S^6$-DAMON's delivered models, we measure their latency on a Google Pixel 3 mobile phone for processing a 10s audio segment with a 16k sampling rate. We also report a real-time factor (RTF) defined as the inference time divided by utterance duration (Gondi, 2022). As shown in Tab. 6, our $S^6$-DAMON can achieve (1) a 1.96× speed-up over PARP with an absolute 2.49% lower WER, and (2) a 1.60× speed-up over the original wav2vec2-base with a comparable WER (+1.03%) or a 2.90× speed-up while maintaining the absolute WER within 10%. This indicates that our method can outperform SOTA ASR pruning methods in real-device efficiency and significantly bridge the gap between speech SSL models and real-time on-device ASR.

**Note that** additional ablation studies can be found in Appendix B-F. These include applying $S^6$-DAMON to noisier datasets and supervised speech models, offering a comprehensive breakdown of each technique's contribution to the reduction in WER, comparing SALAD with other token skipping methods, and validating our design choices.

### 5 CONCLUSION

Both the lack of large-scale transcribed speech data and the prohibitive model complexity hinder ubiquitous ASR systems on mobile platforms. This work develops $S^6$-DAMON to tackle both challenges via effectively sparsifying speech SSL models to enable real-time on-device ASR. Specifically, $S^6$-DAMON integrates SALAD and SAFARI to unlock structured sparsity in both input data and model structures, respectively, where the former exploits and intrinsic redundancy of human utterances and the latter reduces the sparsity discrepancy between finetuning/deployment and enhances the adaptability of sparsity distributions. Extensive experiments validate that $S^6$-DAMON has empowered the deployment of speech SSL models on mobile platforms and our delivered insights could shed light on future innovations in efficiency-oriented speech SSL paradigms.

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

## A OVERVIEW AND OUTLINE

In the appendix, we provide more experiments and analysis as a complement to the main content, which are outlined below:

- We provide a more comprehensive performance breakdown of our $S^6$-DAMON framework as a complement to Tab. 2 of our main text in Sec. B;

- We evaluate the effectiveness of our $S^6$-DAMON on noisy datasets and supervised-trained speech models in Sec. C and Sec. D, respectively;

- We perform more evaluation on our SALAD technique to further validate its promise as a token skipping strategy, including the benchmark with other token skipping methods, validating the necessity of enforcing high SAT recall, and cross-lingual transfer to more languages, in Sec. E;

- We conduct more ablation studies on our SAFARI technique to validate the adopted design choices, including the module-wise sensitivity to structured sparsity and the impact of the adjustment ratio, in Sec. F;

- We introduce the detailed finetuning and real-device measurement settings in Sec. G;

- We discuss the limitations of our framework as well as our future work in Sec. H.

## B BREAKDOWN OF $S^6$-DAMON'S ACHIEVED PERFORMANCE

To demonstrate the contribution of each of our techniques to the final achieved WER reduction, we provide a comprehensive breakdown of $S^6$-DAMON as a complement to the exemplary breakdown in Tab. 2 of our main text. In particular, we finetune wav2vec2-base (Baevski et al., 2020) on LibriSpeech-1h w/ and w/o enabling the proposed semi-supervised distillation on unlabeled LibriSpeech-10h.

**Observation and analysis.** As shown in Tab. 7, we can observe that (1) enabling SAFARI only can outperform the SOTA ASR pruning baseline PARP (Lai et al., 2021), e.g., an absolute 29.89%/14.56% WER reduction for reducing 64.29% FLOPs w/o and w/ distillation, respectively; (2) enabling both SAFARI and SALAD consistently wins a better WER-FLOPs trade-off than enabling SAFARI only, e.g., an absolute 3.06%~4.51% and 2.99%~5.51% lower WER over SAFARI for reducing >50% FLOPs w/o and w/ distillation, respectively; (3) the proposed semi-supervised distillation is generally effective on both our method and the baseline PARP, especially under more stringent low-resource settings. This set of experiments further indicates the importance of input data sparsity as a critical dimension of speech SSL models' redundancy, which is highly desirable to be exploited for pushing forward the achievable WER-FLOPs frontier.

Table 7: Visualize the breakdown of the achieved WER reduction over the baseline PARP (Lai et al., 2021) when finetuned on LibriSpeech-1h. "w/o dist." and "w/ dist." denote without/with distillation, respectively.

| PARP | | | SAFARI (Ours) | | | SAFARI + SALAD (Ours) | | |
|---|---|---|---|---|---|---|---|---|
| FLOPs Savings (%) | w/o dist. | w/ dist. | FLOPs Savings (%) | w/o dist. | w/ dist. | FLOPs Savings (%) | w/o dist. | w/ dist. |
| 27.90% | 23.71 | 20.32 | 27.90% | 22.14 | 19.11 | **40.78%** | 22.56 | **19.17** |
| 41.38% | 34.57 | 29.07 | 41.38% | 26.17 | 20.11 | **46.94%** | 23.33 | **20.16** |
| 53.52% | 50.68 | 35.83 | 53.52% | 29.48 | 23.78 | **57.07%** | 26.42 | **20.79** |
| 64.29% | 64.52 | 44.81 | 64.29% | 34.63 | 30.25 | **66.18%** | 30.12 | **24.74** |

Table 8: Benchmark our $S^6$-DAMON and SAFARI with the baseline PARP (Lai et al., 2021) on the LibriSpeech test-other set with noisy speech when finetuned with different resources.

| Resource | PARP | | SAFARI (Ours) | | S6-DAMON (Ours) | |
|---|---|---|---|---|---|---|
| | FLOPs Savings (%) | WER (%) | FLOPs Savings (%) | WER (%) | FLOPs Savings (%) | WER (%) |
| $D_T$=Libri-1h $D_U$=Libri-10h | 27.90% | 30.09 | 27.90% | 27.13 | **40.78%** | **27.78** |
| | 41.38% | 43.45 | 41.38% | 29.64 | **46.94%** | **28.76** |
| | 53.52% | 51.69 | 53.52% | 34.37 | **57.07%** | **30.77** |
| | 64.29% | 60.69 | 64.29% | 44.61 | **66.18%** | **37.34** |
| $D_T$=Libri-1h $D_U$=Libri-100h | 27.90% | 26.71 | 27.90% | 25.35 | **40.78%** | **25.47** |
| | 41.38% | 33.67 | 41.38% | 26.41 | **46.94%** | **26.46** |
| | 53.52% | 37.86 | 53.52% | 29.63 | **57.07%** | **27.56** |
| | 64.29% | 43.26 | 64.29% | 33.95 | **66.18%** | **31.44** |
| $D_T$=Libri-100h $D_U$=None | 27.90% | 16.17 | 27.90% | 15.53 | **40.78%** | **15.91** |
| | 41.38% | 20.65 | 41.38% | 15.73 | **46.94%** | **15.90** |
| | 53.52% | 27.63 | 53.52% | 18.24 | **57.07%** | **17.26** |
| | 64.29% | 37.60 | 64.29% | 21.47 | **66.18%** | **19.97** |

## C    EVALUATION ON NOISY DATASETS

To validate the robustness of our technique on noisy speech, we further benchmark our $S^6$-DAMON and SAFARI with the SOTA ASR pruning baseline PARP (Lai et al., 2021) on the LibriSpeech test-other set (Panayotov et al., 2015) with background noise when finetuned using different resources.

**Observation and analysis.** As shown in Tab. 8, we can observe that (1) although both our technique and the baseline PARP suffer from a larger WER increase on the noisy test-other set as compared to the reported ones on the test-clean set in our main text, both our $S^6$-DAMON and SAFARI still outperform PARP, e.g., an absolute 17.63%/16.13% lower WER for reducing ¿64% FLOPs when finetuned on LibriSpeech-100h, respectively; (2) our $S^6$-DAMON, i.e., enabling both SAFARI and SALAD, still achieves a better WER-FLOPs trade-off than enabling SAFARI only, e.g., an absolute 7.27% lower WER for reducing ¿64% FLOPs when finetuned on $D_T$=Libri-1h and $D_U$=Libri-10h, indicating the consistent effectiveness of SALAD in terms of further pushing forward the compression frontier on noisy datasets.

Table 9: Benchmark our $S^6$-DAMON and SAFARI with the baseline PARP (Lai et al., 2021) on the supervised-trained wav2vec2-base using different finetuning resources.

| Resource | PARP | | SAFARI (Ours) | | S6-DAMON (Ours) | |
|---|---|---|---|---|---|---|
| | FLOPs Savings (%) | WER (%) | FLOPs Savings (%) | WER (%) | FLOPs Savings (%) | WER (%) |
| Libri-1h | 27.90% | 13.58 | 27.90% | 8.73 | **40.78%** | **8.87** |
| | 41.38% | 29.36 | 41.38% | 13.34 | **46.94%** | **9.90** |
| | 53.52% | 61.92 | 53.52% | 20.73 | **57.07%** | **14.11** |
| Libri-10h | 27.90% | 10.57 | 27.90% | 5.93 | **40.78%** | **5.98** |
| | 41.38% | 14.32 | 41.38% | 7.50 | **46.94%** | **6.65** |
| | 53.52% | 34.83 | 53.52% | 11.55 | **57.07%** | **8.23** |

## D    EVALUATION ON SUPERVISED-TRAINED SPEECH MODELS

Although the main goal of our work is to trim down the complexity of speech SSL models, following the de-facto paradigm for low-resource ASR, our method can be also seamlessly applied to speech models trained in a supervised manner. To demonstrate this, we benchmark our $S^6$-DAMON and SAFARI with the baseline PARP (Lai et al., 2021) on supervised-trained wav2vec2-base using varied finetuning resources with distillation enabled for both our method and PARP. More specifically, we

adopt a sparsity of 0.2/0.3/0.4 for PARP and SAFARI and adopt a $(sr, sp)$ pair, which are defined in Sec. 3.2/3.3 of our main text, respectively, of (0.4, 0.2)/(0.6, 0.2)/(0.6, 0.3) for $S^6$-DAMON.

**Observation and analysis.** As shown in Tab. 9, we can observe that (1) again, both our $S^6$-DAMON and SAFARI can consistently outperform PARP across different data resource settings and FLOPs savings, and (2) our $S^6$-DAMON can achieve a better WER-FLOPs trade-off than SAFARI, indicating the necessity of exploiting input data sparsity via SALAD.

# E  MORE EVALUATION ON THE EFFECTIVENESS OF SALAD

## E.1  BENCHMARK WITH OTHER TOKEN SKIPPING METHODS

To validate the effectiveness of the skipping strategies made by SALAD, we benchmark with three token skipping methods: (1) random skip, (2) uniform skip with a similar effect as reducing the sampling rate, and (3) layer-wise adaptive skip (Wang et al., 2021) based on attention scores, which is initially designed for NLP. In particular, we finetune wav2vec2-base with each of the skipping methods on LibriSpeech-1h and control their skip ratios to ensure comparable FLOPs savings.

Table 10: Benchmark our SALAD with different token skipping strategies.

| Method | FLOPs Saving (%) | WER (%) |
|---|---|---|
| **SALAD (Ours)** | **24.3** | **19.17** |
| | **32.0** | **20.21** |
| Uniform Skip | 20.0 | 19.89 |
| | 30.0 | 24.63 |
| Random Skip | 20.0 | 43.98 |
| | 30.0 | 79.87 |
| Adaptive Skip (Wang et al., 2021) | 15.0 | 25.658 |
| | 22.5 | 36.193 |

**Observation and analysis.** As shown in Tab. 10, we can observe that (1) our SALAD consistently wins the lowest WER under comparable FLOPs, e.g., an absolute 4.42% WER reduction over the strongest baseline uniform skip when saving >30% FLOPs; and (2) the adaptive skip method can hardly surpass the simple uniform skip strategy, indicating that without considering the intrinsic properties of human speech, the monotonic alignment between the input speech and output transcriptions can be easily destroyed.

## E.2  THE NECESSITY OF ENFORCING HIGH RECALL ON SATs

We train two SALADs w/ and w/o recall-aware training (RAT), which are next applied on wav2vec2-base with different NAT $sr$ on LibriSpeech-1h. As shown in Tab. 11, explicitly enforcing a high recall on SATs leads to consistently lower WER especially under larger $sr$, validating the necessity of maximally covering all SATs.

Table 11: Validate the necessity of RAT.

| Setting | w/o RAT | w/ RAT |
|---|---|---|
| Acc (%) | **79.38** | 75.69 |
| Recall (%) | 64.38 | **89.08** |
| NAT $sr$=0.4 | 19.56 | **18.42** |
| NAT $sr$=0.6 | 21.63 | **19.17** |
| NAT $sr$=0.8 | 23.89 | **20.21** |

## E.3  CROSS-LINGUAL TRANSFER OF SALAD

**Setup.** Considering the semi-supervised training scheme of SALAD requires a large set of untranscribed speech, which may not be available for some spoken languages, we evaluate whether the SALAD trained on English can be directly transferred to detect SATs/NATs for other languages. In particular, we transfer SALAD trained on untranscribed LibriSpeech-100h to pursue the input sparsity for finetuning wav2vec2-base on Dutch, Spanish, and Mandarin from CommonVoice (Ardila et al., 2019).

Table 12: Evaluate the phoneme recognition rate (PER) when applying SALAD trained on English to other languages.

| $sr$ | Dutch | Spanish | Mandarin |
|---|---|---|---|
| - | 19.82 | 13.86 | 26.67 |
| 0.2 | 18.89 | 13.76 | 26.61 |
| 0.4 | 19.16 | 13.85 | 26.89 |
| 0.6 | 19.55 | 13.99 | 26.84 |
| 0.8 | 20.09 | 14.32 | 28.46 |

**Observation and analysis.** As shown in Tab. 12, we can see that SALAD trained on English can transfer well to other languages, e.g., achieve a comparable or lower PER under an $sr$ of 0.6, indicating that SALAD can extract general phonetic features that can be shared across spoken languages.

In addition, we also measure the achieved accuracy and recall of detecting SATs achieved by SALAD on ten languages from CommonVoice, each with 1h labeled speech following the data split in (Conneau et al., 2020). As shown in Tab. 13, we can observe that our SALAD can consistently achieve a >68% accuracy and a >85% recall across all languages, indicating the general effectiveness of SALAD across languages.

Table 13: Visualize the accuracy and recall of SALAD on ten languages from CommonVoice (Ardila et al., 2019).

| Language | Dutch | Mandarin | Spanish | Tartar | Russian | Italian | Kyzgyz | Turkish | Swedish | France |
|---|---|---|---|---|---|---|---|---|---|---|
| Accuracy (%) | 70.02 | 74.52 | 69.53 | 73.89 | 77.52 | 76.22 | 68.12 | 71.65 | 68.34 | 74.91 |
| Recall (%) | 91.51 | 85.65 | 95.31 | 89.27 | 85.92 | 86.44 | 94.36 | 88.76 | 94.36 | 87.16 |

Table 14: Apply our SARARI on wav2vec2-base/LibriSpeech-1h with varied FFN/SA sparsity.

| FFN/SA sparsity | GFLOPs | WER (%) | FFN/SA sparsity | GFLOPs | WER (%) | FFN/SA sparsity | GFLOPs | WER (%) |
|---|---|---|---|---|---|---|---|---|
| 0.2/0.2 | 33.024 | 23.19 | 0.3/0.3 | 26.96 | 23.19 | 0.4/0.4 | 21.52 | 31.97 |
| 0.1/0.3 | 33.96 | **22.13** | 0.2/0.4 | 27.61 | **24.52** | 0.3/0.5 | 21.89 | **29.72** |
| 0.3/0.1 | 32.016 | 25.19 | 0.4/0.2 | 26.23 | 31.05 | 0.5/0.3 | 21.08 | 47.77 |

# F  ABLATION STUDIES ON THE DESIGN CHOICES OF SAFARI

## F.1  MODULE-WISE SENSITIVITY TO STRUCTURED SPARSITY

To justify our choice of module-wise sparsity distribution, we conduct an ablation study to apply our SAFARI on wav2vec2-base on top of LibriSpeech-1h and vary the sparsity in FFN and SA under comparable FLOPs. As shown in Tab. 14, we consistently find that FFNs are more sensitive to structured pruning, especially under large sparsity. This may be because task-specific information is mostly learned by FFNs during finetuning thus their sufficient complexity is crucial. Therefore, we by default set their sparsity to satisfy $(sp_{SA} + sp_{FFN})/2 = sp$ and $sp_{SA} - sp_{FFN} = 0.2$ for a given sparsity $sp$ in Sec. 4 of our main text.

## F.2  THE CHOICE OF ADJUSTMENT RATIOS

We vary the adjustment ratio $ar$ under different sparsity $sp$ on top of wav2vec2-base and LibriSpeech-1h. As shown in Tab. 15, we observe that the optimal $ar$ varies for different $sp$ and in general larger sparsity calls for higher adaptability of sparsity distributions. Therefore, we set $ar = sp$ across all experiments in Sec. 4 of our main text.

Table 15: The achieved WER under varied adjustment ratios $ar$ and sparsity $sp$.

| $ar$ / $sp$ | 0.2 | 0.3 | 0.4 |
|---|---|---|---|
| 0.1 | 23.75 | 26.53 | 33.14 |
| 0.2 | **22.86** | 28.65 | 31.97 |
| 0.3 | 22.96 | **23.19** | 30.76 |
| 0.4 | 23.18 | 26.79 | **30.74** |

# G  MORE DETAILS ABOUT EXPERIMENT SETUP

**Finetuning settings.** We implement S[6]-DAMON on top of fairseq (Ott et al., 2019) and we follow the default finetuning settings for each task, i.e., the default configurations in fairseq for ASR/PR and those in SUPERB (Yang et al., 2021) for other speech processing tasks. In particular, all experiments on ASR/PR are trained for 12k/15k/20k/80k steps on the 10m/1h/10h/100h splits using an Adam optimizer with an initial learning rate of 5e-5 plus a tri-stage schedule (Baevski et al., 2020). We do not freeze all the transformer layers for the first 10k steps (Baevski et al., 2020), following (Lai et al., 2021). All experiments are trained on two NVIDIA A5000 GPUs using a distributed data-parallel scheme. In addition, considering the transformer backbone accounts for >90% parameters in the speech SSL models, all the reported FLOPs/Params savings and sparsity ratios are relative to the transformer, following PARP (Lai et al., 2021).

**Measurement settings.** For the measurement on the Google Pixel 3 mobile phone, all Pytorch models are converted to ONNX and then compiled to the TFLite format, following (Li et al., 2021a). We separately compile (a) the convolutional feature extractor + SALAD, and (b) the transformer backbone, where the output of (a) is fed into (b) as its input. The latency on both (a) and (b) as well as the overall speed-up are reported in Sec. 4.4 of our main text.

# H  LIMITATIONS AND FUTURE WORK

Our work has two limitations: (1) we only apply our technique on the transformer component of speech SSL models while leaving the convolutional feature extractor that transforms raw audio

into tokens untouched since we find that it is sensitive to compression and suffers from non-trivial WER increases under larger structured sparsity. Although the transformer component dominates the computational cost, the overall speed-up can be limited under extremely large sparsity, e.g., our method can achieve $8.15\times$ speed-up on the transformer component while the overall (i.e., whole-model) speed-up is constrained to $2.90\times$ according to Tab. 6 of our main text. In our future work, we will design more lightweight and robust feature extractors, either in the frequency domain or in the time domain, to push forward the achievable overall speed-up; (2) although the ultimate goal and potential impacts mentioned in Sec. 1 of our main text is to facilitate the on-device deployment of emerging foundation models (Bommasani et al., 2021), in this work we consider the speech domain first as it is one of the most commonly adopted input modalities on mobile devices. In our future work, we will extend our framework and insights to more advanced foundation models across NLP and CV domains with the goal of democratizing the power of cutting-edge AI foundation models to everyday devices.

