# OpenReview forum: "S$^6$-DAMON: Unlocking Structured Sparsity in Self-Supervised Speech Models via Data-Model Co-Compression"
_ICLR.cc/2024/Conference — ICLR 2024 Conference Withdrawn Submission_

### Official Review · Reviewer_FqdU · 2023-10-19

**Soundness:** 4 excellent
**Presentation:** 1 poor
**Contribution:** 3 good
**Rating:** 3
**Confidence:** 5

**Summary:**

This work proposes two methods that aim at reducing the inference footprint (compute and latency) of ASR (or others, but the paper focuses on ASR) systems based on pre-trained backbones (typical SSL models). SSL backbones are often large transformers (100M - 3B parameters) and dense, resulting in clear challenges when deployed on constrained devices. This article proposes to combine time decimation and pruning to achieve this goal (named data-model co-compression). Time decimation is achieved via a newly introduced model that basically predicts the first acoustic frame (or acoustic vector) given an ensemble of repeated labels (hence zeroing out all the subsequent ones). Structured pruning is achieved by modifying an existing method, PARP, slightly. The results clearly support the effectiveness of this combination.

**Strengths:**

This work establishes, via extensive experiments, an important demonstration that is often being done behind closed gates in the industry: an SSL-based ASR system can be pruned and achieve good performance. The two methods aren't specifically novel or groundbreaking, but they work, and explicit a direction that could help researchers and engineers to deploy such models on devices. **The question addressed in this paper, i.e. how to make it feasible to use SSL models on constrained devices, is of utmost interest and extremely timely**. I do believe that this paper **could** really interest the community and raise a lot of interest, however, it currently suffers from many other weaknesses that prevent it from being publishable in ICLR.

**Weaknesses:**

Two main issues arise from the current version of the work:
1. The first part (up to the model definition) tries to oversell the idea too much i.e. write a lot about a few key points.
2. The second part of the paper (model definition to conclusion) squeezes an absurd amount of information in not a lot of space.

Again, the idea is extremely simple, SSL models are big and costly to infer with, and existing pruning strategies do not work, this work solves the issue. Yes, we end up with many reformulations of the contributions, of the problem, or of the context. This writing style ends up introducing many vague statements such as the fact that big data is a problem for ubiquitous on-device ASR systems, which is not true, actually not really connected. Other examples are "recent breakthroughs" with a paper from 2014 - this is 10 years old. Or the introduction of different names for the two techniques, with different catchy acronyms - be more factual and simple, this is just time-decimation with a modified PARP. An example of the consequence of this writing style is, for instance, that at the end of the introduction, we still don't know if we are talking about reducing the inference cost once deployed (i.e. no personalization) or if it's for on-device fine-tuning. Other vague statements include: "under the SOTA pretrain-and-finetune paradigm, the **most useful features** are learned during the SSL stage" - this is at best not precise and at worst plainly wrong. Or "transformers are often adopted in SOTA speech SSL models to ensure **effective representation learning**" - again, without any reference, this claim is vague - the reality being that people are using transformers models in SSL for speech just because wav2vec 2.0 did it and because all available implementations are using it. We have no idea if a big CNN wouldn't be a more effective model. The introduction, again for example, has two paragraphs that summarize the contributions (basically the same) - why? The "on the data side" and "on the model side" paragraphs are also quite confusing as they try to overcomplexify the definition of each method, which are very simple, but without introducing the formal definition. For instance, the introduction of SAT and NAT basically is unnecessary - this is just a dynamic time decimation based on the implicit language model of the fine-tuned ASR teacher i.e. predicting the first token (or n-first) of the CTC graph. Simplifying this would give much more space to describe properly the difference between this approach and all the other existing time downsampling techniques. It remains unclear to me if a simple 2D convolution with a 2x time reduction (to make the 50hz of wav2vec 2.0 match the usual 25hz of most ASR systems), like most acoustic feature extractors have, wouldn't do the job as well as SALAD. This kind of discussion would be much more relevant than rephrasing the merging of consecutive tokens.

Following this lack of clarity in the context of this article, the related work is an excellent example of the problem: it adds nothing to the article. The related work on ASR and SSL is not necessary for this paper, however, a proper discussion of ASR pruning and time downsampling and how the introduced techniques differ would be highly relevant. As of now, it is absolutely unclear to the reader how the related work relates to the work. Figure 2 (c), due to its size and how it's connected to the text, is extremely hard to follow. It should be a single figure, within a proper section.

The experimental section is way too dense - it somehow feels like many reviewers asked to add many different tasks and they feel completely unconnected. While they all show that the method is better (great) they also completely lose the reader. As a result, a clear narrative is missing and graphs and tables are way too small (asking the reader to zoom in definitely does not help).  For instance, it would make the paper clearer if the Du (unsupervised) variable was not considered. This variable is not exploited in the text and does not bring much to the narrative. It would be better to right a less dense experiment section, but with a proper narrative, rather than having dozens of tables of different experiments packed. All these experiments are great and validate the idea, but they could go in the supplementary material with the other dozens of experiments that are already there ... i.e. that is WAY too much for a simple conference paper.

I believe that too many changes are required to make the paper easy to follow and adapted for publication in this venue, and I would also mention that such a work would be an absolute fit to some journal out there, as it would be given enough space to detail everything and truly make an impact in the community.

**Questions:**

- Why do we care about the "limited transcribed speech", this sounds orthogonal to the problematic (compute efficiency)? I.e. complexifying the narrative for nothing.
- SALAD, unlike convolutional downsampling, will be biased towards the implicit LM of the teacher AND of the one obtained after fine-tuning on the semi-supervised dataset. What if there is an important domain mismatch (at the acoustic level) between the SALAD training and the fine-tuning (which is the case in a lost of SSL real-world use case)? Is SALAD adapted when fine-tuning at the pruning stage?
- The paper claims that the time-downsampling leveraged the intrinsic redundancy in human utterances (duration of the phoneme and the pauses between the phonemes). The paper does not present anything going in that direction - why are phonemes put in the middle here?

---

### Official Review · Reviewer_7VsP · 2023-10-31

**Soundness:** 2 fair
**Presentation:** 2 fair
**Contribution:** 2 fair
**Rating:** 3
**Confidence:** 5

**Summary:**

Authors have proposed a data-model co-compression framework for self-supervised learning for on device ASR models. The method relies on distillation from large models to learn low-footprint models.

**Strengths:**

The paper is nicely written and easy to follow. The approach presented is simple but also intuitive and is being used in practice in one or the other form.

**Weaknesses:**

The novelty of the work is limited.
There are many approaches that exploit intrinsic redundancy in human speech. In the context of ASR, authors claim this as a novelty since perhaps this has not been tried out for Transformers, especially for creating input tokens. However, this is not the case.
SALAD
--> It is not clear what's the difference between CNN layers at input and SALAD? Why additional layers are required? A crude approximation can be achieved using VAD.
--> If compression is the main goal, then pretrained Encodec, like Soundstream, can be explored.
SAFARI
The approach is intuitive, and the adaptive parameter selection is similar to Gided-dropout, ITPS or other sparse training methods inspired by gradient-based sensitivity analysis for running.
https://arxiv.org/abs/2306.12026
https://arxiv.org/pdf/2306.14775.pdf

Code is not made public, making it difficult for readers to replicate the experiments.
https://ieeexplore.ieee.org/document/9916206

Experiments:
Baseline models are poor. WER is quite high.
Why results are only reported on test clean?

**Questions:**

Please see the review.

---

### Official Review · Reviewer_NUX3 · 2023-11-01

**Soundness:** 2 fair
**Presentation:** 2 fair
**Contribution:** 3 good
**Rating:** 5
**Confidence:** 3

**Summary:**

How to deploy the continually growing best neural network based models under the constraint of limited resources of mobile devices?
The present paper addresses this question and investigates ways to streamline the complexity of current speech models.
Specifically, the paper proposes S6-DAMON based on SALAD and SAFARI, which are two methods to unlock sparsity in both input data (by skipping audio tokens) and model structures (by structured sparsity).
In addition to ablations, FLOPs comparison, etc., the paper reports latency and RTF of compressed models on a real device. For example, two operating points of S6-DAMON include:
- 1.5x overall speed up at 2/3 parameters and 1.1x WER.
- 2x overall speed up at 1/2 parameters and 1.5x WER.

**Strengths:**

* The deployment of state-of-the-art models is a timely topic and of high relevance.
* S6-DAMON is a concrete and effective contribution to close this gap. See selected datapoints in the summary above.
* Real-device measurement of S6-DAMON with latency/RTF, in addition to FLOPs comparisons.
* Related work: It's nice to see also early work covered.

**Weaknesses:**

* I find the storyline of the paper with interleaving SSL and post-hoc model compression confusing. My naive assumption is that these are two largely orthogonal topics. The additional results on supervised-trained speech models in Appendix D seem to confirm this assumption. Focusing on model compression applied to different model variants (model size, how the model is trained, ...) would make the paper much simpler and clearer in my opinion. See also Question 1 below.

* Model compression is a typical trade off between accuracy and speed/size. A 2d scatter plot is an effective way to show where a new data point is regarding existing data points and Pareto optimality. I'm missing such a visualization in this paper.

**Questions:**

1. How does SSL affect SALAD/SAFARI, except maybe for the fact that models tend to be even larger for SSL?

2. I understand that imposing sparsity or model compression from scratch is challenging. But adding a few more baseline results for comparison would strengthen the need for rather sophisticated post-hoc model compression methods. For example,

    - How does a reduced frame rate used from scratch compare with skipping tokens as in SALAD?
    - How do smaller models (e.g., fewer nodes per layer for comparison with structured sparsity) trained from scratch compare with S6-DAMON?

3. Table 6: The max. overall speed up is 4 for optimizing the transformer only. The best achieved speed up is 3. I.e., the bottleneck now is “Conv FE”.  Has "Conv FE" already been optimized? How can the model be optimized further? A compression of 50% and a 2x speed up are a non-trivial achievement, but most likely not enough to close the gap.

---

### Official Review · Reviewer_NFEu · 2023-11-06

**Soundness:** 2 fair
**Presentation:** 3 good
**Contribution:** 3 good
**Rating:** 5
**Confidence:** 4

**Summary:**

This paper tackles the problem of on-device deployment of SSL speech models for ASR by proposing two approaches. The first one, called 'SALAD', performs input reduction by skipping non-sailent frames, which are predicted by a binary classifier trained with pseudo labels generated by fine-tuned SSL model. The second one, called  'SAFARI', performs structured model pruning by iterative fine-tuning and re-pruning under target sparsity. The proposed approaches are experimentally evaluated on Librispeech for ASR and SUPERB for other speech processing tasks. Their effectiveness are demonstrated by major results such as WER over different FLOPs/Sparsity and efficiency on a Google Pixel 3 mobile phone.

**Strengths:**

The proposed method for input reduction by skipping non-sailent frames seems to be original and effective for large computation reduction, which can be additionally applied with different model compression methods.
Various experiments covering different tasks, data amount, SSL models and ablations are plus points.

**Weaknesses:**

The major weakness lies in the proposed structured pruning 'SAFARI' and its comparison with the baseline method PARP (claimed SOTA ASR pruning method here), which was proposed for unstructured pruning in 2021. To extend PARP to structured pruning, some naive extension w/o giving details is applied in Sec. 3.3, which shows large degradation compared to the original approach. Most results in Sec. 4 are not clearly stated whether the comparison is done against the original unstructured PARP or the naively-extended structured PARP. The former would make efficiency comparison not fair and the latter would make WER comparison not fair. Actually, the claimed discrepency issue of PARP in Sec. 3.3 seems to be simply tacklable by applying the step 2 of 'SAFARI' to the final iteration of PARP, which may directly yield a better baseline. And in this case, the proposed 'SAFARI' becomes a minor extension of PARP except for the different re-pruning using 'prune-and-grow' (step 3), which is not theoretically nor experimentally shown to be any better in this paper. For structured pruning, the criterion for prune (smallest L1-norm of weight vectors) and for grow (largest L1-norm of weight vectors' gradient) may also introduce discrepency as they do not directly account for the amount of contribution to the output.
After all, it would be most straightforward to directly compare with (SOTA) strucured pruning method, such as Jiang et al. "Accurate and Structured Pruning for Efficient Automatic Speech Recognition", which shows much less WER degradation up to 50% sparsity (also much better than PARP).

Side remark: the method naming ('SALAD', 'SAFARI', 'S^6 DAMON) seems a bit over-done / far-fetched.

**Questions:**

Some mistakes:
- Eq.1: I believe theta_s should be trained to minimize the loss rather than maximize. Also this expression seems incomplete, e.g. if it's a training objective, just be consistent as Eq. 2
- Sec.2 1st paragraph:  sequence-to-sequence modeling covers all CTC, RNN-T and AED models, not only AED.